# Influence of Dispersed TiO_2_ Nanoparticles via Steric Interaction on the Antifouling Performance of PVDF/TiO_2_ Composite Membranes

**DOI:** 10.3390/membranes12111118

**Published:** 2022-11-09

**Authors:** Jie Zhang, Zicong Jian, Minmin Jiang, Bo Peng, Yuanyuan Zhang, Zhichao Wu, Junjian Zheng

**Affiliations:** 1School of Chemistry and Materials Engineering, Huizhou University, 46 Yanda Road, Huizhou 516007, China; 2College of Life and Environmental Science, Guilin University of Electronic Technology, 1 Jinji Road, Guilin 541004, China; 3School of Environmental Science and Engineering, South University of Science and Technology of China, No. 1088 Xueyuan Avenue, Shenzhen 518055, China; 4Department of Electrical Engineering, National Cheng Kung University, No. 1 Daxue Road, Tainan 701401, China; 5State Key Laboratory of Pollution Control and Resource Reuse, School of Environmental Science and Engineering, Tongji University, 1239 Siping Road, Shanghai 200092, China

**Keywords:** TiO_2_ nanoparticles, dispersion, steric interaction, composite membrane, antifouling performance

## Abstract

Herein, the influence of various contents of polyethylene glycol (PEG) on the dispersion of TiO_2_ nanoparticles and the comprehensive properties of PVDF/TiO_2_ composite membranes via the steric hindrance interaction was systematically explored. Hydrophilic PEG was employed as a dispersing surfactant of TiO_2_ nanoparticles in the pre-dispersion process and as a pore-forming additive in the following membrane preparation process. The slight overlap shown in the TEM image and low TSI value (<1) of the composite casting solution indicated the effective dispersion and stabilization under the steric interaction with a PEG content of 6 wt.%. Properties such as the surface pore size, the development of finger-like structures, permeability, hydrophilicity and Zeta potential were obviously enhanced. The improved antifouling performance between the membrane surface and foulants was corroborated by less negative free energy of adhesion (about −42.87 mJ/m^2^), a higher interaction energy barrier (0.65 KT) and low flux declination during the filtration process. The high critical flux and low fouling rate both in winter and summer as well as the long-term running operation in A/O-MBR firmly supported the elevated antifouling performance, which implies a promising application in the municipal sewage treatment field.

## 1. Introduction

The membrane bioreactor (MBR) has received considerable attention in the field of wastewater treatment because of the high effluent quality, small footprint, low biomass production, complete separation of hydraulic retention time (HRT) and solids retention time (SRT) [1]. Microfiltration (MF) and ultrafiltration (UF) processes used in MBR have critically been deteriorated by membrane fouling, which leads to the significant decline of filtration performance, high energy consumption, frequent washing processes and the replacement of membranes, etc. [2] To elucidate the fouling behaviors of membranes in MBR, countless studies have been conducted, such as membrane modification, operating condition optimization, and so on [3]. Among them, membrane modification by incorporating the original membrane material with nanoparticles has been identified as a straightforward way to deal with the thorny membrane fouling issue [4].

As an environmentally friendly and chemically stable material, TiO_2_ nanoparticles have been constantly used to modify the antifouling performance of the original polymeric membrane owing to their excellent hydrophilicity and self-cleaning capability [5]. The preparation of a polymer/TiO_2_ composite membrane typically involves the coating of TiO_2_ nanoparticles on the surface of a membrane or blending TiO_2_ nanoparticles into the casting solution [6]. Considering the unstable coating modification versus operation time, the blending modification has been proven to be a facile, effective and stable method to combine the comprehensive characteristics of nanoparticles and pristine polymer in a composite membrane. Traditionally, TiO_2_ nanoparticles were directly dispersed with polymer in a casting solution to form a composite membrane by the phase inversion procedure [7]. However, the comprehensive performance modification of the polymer/TiO_2_ composite membrane, including permeability and the antifouling property, has been proven to be unfortunately affected by the unavoidable aggregation of TiO_2_ nanoparticles via the crude addition in the casting solution [8]. A novel convenient method should be developed to bring down the negative effects of the agglomeration of TiO_2_ nanoparticles on the membrane modification efficiency.

Physical surface treatment has been reported to reduce the agglomeration phenomenon by lowering the surface energy of nanoparticles via steric and/or electrostatic mechanisms [9]. PEG, known as a nonionic surfactant of nanoparticles and a hydrophilic pore-forming additive, plays a major role in the field of membrane fabrication. The steric interaction of the hydrophilic chains of PEG assist in the stable dispersion of nanoparticles in solution [10]. In our previous study, PEG was proved to contribute the dispersion in composite casting solutions by the inner steric interaction mechanism during the blending process and also benefited the fabrication of a uniform polymer/TiO_2_ composite membrane in the phase inversion process [11]. Well-dispersed TiO_2_ nanoparticles under the steric interaction of PEG were precisely proved to modify the stability of the composite casting solution as well as the properties of the composite membrane, such as hydrophilicity, permeability and antifouling performance, etc. In view of the detrimental influence of superfluous nanoparticles in the sewage treatment system, the dispersion mechanism of a small amount of TiO_2_ nanoparticles by varying the content of an organic surfactant deserves a clear consideration in the antifouling performance modification of the polymer/TiO_2_ composite membrane.

In this study, poly(vinylidene fluoride) (PVDF) was taken as the pristine polymer material for membrane fabrication due to its outstanding mechanical property, high chemical resistance, excellent thermal stability, and so on [12]. The effects of various contents of PEG on the dispersion of TiO_2_ nanoparticles as a surfactant and the subsequent membrane fabrication process as a pore-forming additive were investigated. The dispersion of the TiO_2_ nanoparticles in organic suspensions and composite casting solutions were studied by employing transmission electron microscopy (TEM) and multiple light scattering spectroscopy (MLiSSP), respectively. The morphologies of the PVDF/TiO_2_ composite membranes fabricated by various contents of PEG were characterized by scanning electron microscopy (SEM). The surface roughness and 3D morphologies were examined by atomic force microscopy (AFM). Permeabilities, hydrophilicities and mechanical properties were also investigated. The antifouling performance of PVDF/TiO_2_ composite membranes was theoretically simulated by the extended Derjaguin–Landau–Verwey–Overbeek (XDLVO) theory with soluble microbial products (SMP) as foulants. In addition, SMPs containing polysaccharides, protein, humic acid and other polymeric compounds are excreted into solution during substrate metabolism, biomass growth, biomass decay, etc., and have been identified as significant biological foulants [13]. The antifouling performance of various PVDF/TiO_2_ composite membranes for SMPs was also experimentally explored by the batch filtration procedure. A pilot-scale submerged anoxic/oxic membrane bioreactor (A/O-MBR) mentioned in our previous study was utilized to evaluate the antifouling performance for membranes when targeting the municipal wastewater treatment application [14]. The critical flux and fouling rate in the A/O-MBR device were separately examined in both winter and summer. To further validate the antifouling performance of various PVDF/TiO_2_ composite membranes, a long-term running process in an actual municipal wastewater treatment system was also conducted.

## 2. Materials and Methods

### 2.1. Materials

PVDF with an average molecular weight (Mw) of 670~700 kDa was used as the polymer material. Dimethylsulfoxide (DMSO) was used as the organic solvent in this study. PEG with an average Mw of 400 Da was simultaneously used as the dispersing surfactant of nanoparticles and the pore-forming additive during the membrane fabrication process. TiO_2_ nanoparticles with an average particle size of 21 nm were obtained from Sigma-Aldrich (St. Louis, MO, USA). SMPs used as the model foulants in this study were obtained by filtrating sludge in an aerobic zone. An SMP solution at pH 7.0 was adjusted by 0.1 M NaOH and 0.1 M HCl. Deionized water was used throughout this study.

### 2.2. Membrane Preparation

PVDF/TiO_2_ composite membranes with a TiO_2_ nanoparticle concentration of 0.15 wt.% were prepared by phase inversion via the immersion precipitation method. PEG concentrations were set as 0 wt.%, 2 wt.%, 4 wt.%, 6 wt.% and 8 wt.%, respectively, and the PVDF/TiO_2_ composite membranes were correspondingly termed as E1, E2, E3, E4 and E5. The first 2/3 solvent (about 60 vol.%) was used to dissolve the PVDF at 80 °C for 4 d, and homogenous polymeric solutions were prepared during this process. The remainder of the solvent was employed to jointly disperse the TiO_2_ nanoparticles with various contents of PEG. Suspensions of TiO_2_ nanoparticles were subjected to ultrasonication at 20 °C for 20 min and then blended with the pre-prepared homogenous polymeric solutions. The final casting solutions for the PVDF/TiO_2_ composite membranes E1–E5 were attained after stirring for another 3 d at 80 °C. The casting solutions were subsequently casted on porous polyester non-woven fabrics/flat glass plates with a scraper clearance of 250 μm. After being briefly (30 s) evaporated into the ambient air, these casting films together with fabrics/flat glass plates were submerged into the coagulation bath (deionized water) at room temperature to form the porous PVDF/TiO_2_ composite membranes E1–E5.

### 2.3. Dispersion Analysis of TiO_2_ Nanoparticles

The dispersion of TiO_2_ nanoparticles in suspension was observed by employing TEM (JEM-2100, JEOL, Tokyo, Japan). The stability implying the kinetics mechanism of TiO_2_–PEG in nano-composited casting solutions was studied by MLiSSP (Turbiscan Tower, Formulaction, Toulouse, France) with a near-infrared light source (λ = 880 nm) at 80 °C for 100 min. Transmission (*T*) and backscattering (*BS*) signals were monitored by two detector devices, i.e., the transmission detector and the backscattering detector along the cell height throughout the measurement process [15]. A statistical factor, Turbiscan Stability Index (*TSI*), indicating the stability of different casting solution samples can be obtained using Equation (1). A higher *TSI* value indicates a lower stability of the given composite casting solution system [16].
(1)$$TSI=\sqrt{\frac{\sum_{i=1}^{n}{\left(x_{i}-x_{BS}\right)}^{2}}{n-1}}$$
where *x_i_* is the average backscattering for each minute of measurement, *x_BS_* refers to the average *x_i_* and *n* is the number of scans.

### 2.4. Membrane Characterization

The surface morphologies and cross-section morphologies of PVDF/TiO_2_ composite membranes E1–E5 were observed by field emission scanning electron microscopy (FESEM, S-4800, Hitachi, Tokyo, Japan). The average pore size on the surfaces of PVDF/TiO_2_ composite membranes E1–E5 was statistically collected by Image-Pro Plus 6.0 software (Media Cybernetics, Rockville, MD, USA).

Membrane porosity was determined by applying Equation (2) according to the gravimetric method [17].
(2)$$\varepsilon =\frac{m_{1}-m_{2}}{\rho_{w}\cdot A\cdot l}$$
where *m*_1_ and *m*_2_ refer to the weights of the wet and dry membranes (g), respectively. *ρ_w_* is the water density (1 g/cm^3^) and *A* is the effective area of the membrane (cm^2^). Membrane thickness *l* (cm) was measured by a micrometer caliper at different areas of membrane surfaces three times.

Pure water flux reflecting water permeability was obtained by filtrating membranes at a trans-membrane pressure (TMP) of 30 kPa in a dead-end filtration cell and calculated by Equation (3) [18].
(3)$$J=\frac{m}{A\cdot \Delta t}$$
in which *m*, *A* and Δ*t* refer to the volume of permeated water (L), the effective membrane filtration area (m^2^) and the permeation time (h), respectively.

Mechanical properties of PVDF/TiO_2_ composite membranes E1–E5 were tested by a microcomputer-controlled electric universal testing machine (Sans Material Testing Corporation, Shenzhen, China) at room temperature. The attenuated total reflectance–Fourier transform infrared (ATR–FTIR, Thermo Electron Corporation, Waltham, MA, USA) was used to analyze the functional groups presented on the surfaces of PVDF/TiO_2_ composite membranes E1–E5. The surface roughness of each membrane sample, characterized by three-dimensional (3D) morphology images, average roughness (*R*a) and root-mean-square roughness (*R*q) was determined by AFM (Dimension 5000, Bruker AXS, Santa Barbara, CA, USA). The contact angle characterizing membrane surface hydrophilicity was observed by an optical measurement system (OCA 15 Plus, Data physics GmbH, Filderstadt, Germany). The contact angle of the SMP sample was obtained by observing a reverse osmosis (RO) membrane with the interception of the SMP on the surface. The Zeta potential of the membrane surface was measured by a streaming potential analyzer (EKA 1.00, Anton-Paar, Graz, Swiss). A 10 mM KCl solution with a pH value of 7.0 was used as the flowing liquid when measuring the streaming potential of PVDF/TiO_2_ composite membranes E1–E5. The average size and Zeta potential of the SMP sample were determined by a Zetasizer analyzer (Nano-ZS90, Malvern Instruments, Malvern, UK).

### 2.5. Evaluation of Membrane Antifouling Performance

#### 2.5.1. XDLVO Theory Analysis

Membrane surface tension parameters can be obtained using extended Young’s Equations (4)–(6) with contact angles, which were determined by applying three probe liquids (water, formamide and diiodomethane) [19].
(4)$$\gamma^{\mathrm{AB}}=2\sqrt{\gamma^{+}\gamma^{-}}$$
(5)$$\gamma^{\mathrm{TOT}}=\gamma^{\mathrm{LW}}+\gamma^{\mathrm{AB}}$$
(6)$$\left(1+\cos\theta \right)\;\gamma_{l}{}^{\mathrm{TOT}}=2\left(\sqrt{\gamma_{s}{}^{\mathrm{LW}}\gamma_{l}{}^{\mathrm{LW}}}+\sqrt{\gamma_{s}{}^{+}\gamma_{l}{}^{-}}+\sqrt{\gamma_{l}{}^{+}\gamma_{s}{}^{-}}\right)$$
in which *γ*^+^ is the electron acceptor parameter, *γ*^−^ the electron donor parameter, *γ*^AB^ the acid–base (AB) component of surface tension, *γ*^LW^ the Liftshiz–van der Waals (LW) component of surface tension, *γ*^TOT^ the total surface tension and *θ* the contact angle. The subscript (s) means either the membrane surface or foulants (SMP in this study) and (l) denotes the probe liquid used in the measurements.

The free energy of adhesion between the membranes and SMP per unit area can be worked out by utilizing Equation (7). The LW, AB and electrostatic (EL) interaction free energy components at the minimum separation distance *h*_0_ (*h*_0_ ≈ 0.158 nm), i.e., Δ*G_h_*_0_^LW^, Δ*G_h_*_0_^AB^ and Δ*G_h_*_0_^EL^ can be determined by Equations (8)–(10), respectively [20].
(7)$$\Delta G_{h_{0}}^{\mathrm{TOT}}=\Delta G_{h_{0}}^{LW}+\Delta G_{h_{0}}^{AB}+\Delta G_{h_{0}}^{EL}$$
(8)$$\Delta G_{h_{0}}^{LW}=2\left(\sqrt{\gamma_{l}{}^{\mathrm{LW}}}-\sqrt{\gamma_{m}{}^{\mathrm{LW}}}\right)\left(\sqrt{\gamma_{c}{}^{\mathrm{LW}}}-\sqrt{\gamma_{l}{}^{\mathrm{LW}}}\right)$$
(9)$$\Delta G_{h_{0}}^{AB}=2\sqrt{\gamma_{l}{}^{+}}\left(\sqrt{\gamma_{m}{}^{-}}+\sqrt{\gamma_{c}{}^{-}}-\sqrt{\gamma_{l}{}^{-}}\right)+2\sqrt{\gamma_{l}{}^{-}}\left(\sqrt{\gamma_{m}{}^{+}}+\sqrt{\gamma_{c}{}^{+}}-\sqrt{\gamma_{l}{}^{+}}\right)-2\left(\sqrt{\gamma_{m}{}^{+}\gamma_{c}{}^{-}}+\sqrt{\gamma_{c}{}^{+}\gamma_{m}{}^{-}}\right)$$
(10)$$\Delta G_{h_{0}}^{\mathrm{EL}}=\frac{\kappa \varepsilon_{0}\varepsilon_{r}}{2}\left(\zeta_{c}^{2}+\zeta_{m}^{2}\right)\times (1-\coth(\kappa h_{0})+\frac{2\zeta_{m}\zeta_{c}}{\left(\zeta_{c}^{2}+\zeta_{m}^{2}\right)}\csc h\left(\kappa h_{0}\right))$$
where *ε*_0_ is the dielectric permittivity of a vacuum, *ε*_r_ is the dielectric constant of water, *ε*_0_*ε*_r_ is the dielectric permittivity of the fluid, *ζ*_c_ is Zeta potential of the SMP solution, *ζ*_m_ is the surface potential of the membrane and *κ* is the inverse Debye screening length. The subscripts m, l and c refer to the membrane, bulk liquid (water in this study) and colloidal foulants (SMP in this study), respectively. The free energy of cohesion for the membranes and the corresponding components can also be obtained by Equations (8)–(10), when *γ*_c_ is replaced by *γ*_m_. The inverse Debye screening length, *κ*, is determined by Equation (11) [21].
(11)$$\kappa =\sqrt{\frac{e^{2}\sum n_{i}z_{i}^{2}}{\varepsilon_{r}\varepsilon_{0}kT}}$$
where *e* is the electron charge, *n_i_* is the number concentration of ion *i* in the bulk solution, *z_i_* is the valence of ion *i*, *k* is Boltzmann’s constant and *T* is the absolute temperature.

The total energy balance for aqueous systems relates to LW, AB and the EL interaction energy (shown as Equation (12)). The LW, AB and EL interaction energy components between the membrane and SMP (*U*_mlc_^LW^, *U*_mlc_^AB^ and *U*_mlc_^EL^) can be calculated through Equations (13)–(15), respectively [22].
(12)$$U_{\mathrm{mlc}}{}^{\mathrm{XDLVO}}=U_{\mathrm{mlc}}{}^{\mathrm{LW}}+U_{\mathrm{mlc}}{}^{\mathrm{EL}}+U_{\mathrm{mlc}}{}^{\mathrm{AB}}$$
(13)$$U_{\mathrm{mlc}}{}^{\mathrm{LW}}=2\pi \Delta G_{h_{0}}^{LW}\frac{h_{0}^{2}a}{h}$$
(14)$$U_{\mathrm{mlc}}{}^{\mathrm{AB}}=2\pi a\lambda \Delta G_{h_{0}}^{AB}\exp\left[\frac{h_{0}^{}-h}{h}\right]$$
(15)$$U_{\mathrm{mlc}}{}^{\mathrm{EL}}=\pi \varepsilon_{0}\varepsilon_{r}a\left[2\zeta_{c}\zeta_{m}\ln\left(\frac{1+e^{-\kappa h}}{1-e^{-\kappa h}}\right)+\left(\zeta_{c}^{2}+\zeta_{m}^{2}\right)\ln(1-e^{-2\kappa h})\right]$$
where *a* is the radius of foulants (SMP in the study), *h* the separation distance between membrane and foulants, and *λ* is the decay length of AB interactions (0.6 nm).

#### 2.5.2. SMP Filtration

The antifouling propensity for SMP samples on PVDF/TiO_2_ composite membranes E1–E5 was determined using a filtration cell (MSC300, Mosu Corporation, Shanghai, China) at room temperature under a magnetic stirring rate of 500 rpm. Prior to the filtration process, the membrane sample was soaked in DI water for at least 24 h to remove surface impurities and then pre-compressed by filtrating DI water for 30 min at 0.03 MPa. The SMP sample was also filtrated by an RO membrane in the cell in order to obtain the SMP-covered membrane for determining the contact angle value.

#### 2.5.3. Critical Flux

The critical flux of the membranes was determined in the pilot-scale submerged A/O-MBR located in the Quyang municipal wastewater treatment plant of Shanghai. The step duration and incremental flux were set as 15 min and 3 L/(m^2^·h), separately. The critical flux of PVDF/TiO_2_ composite membranes E1–E5 was measured both in winter and summer. When conducting the critical flux determination experiment, the aqueous temperature in A/O-MBR was recorded as 14 °C in winter and 24 °C in summer. Considering the negative effect of the aqueous temperature in A/O-MBR on membrane fouling behaviors, the initial flux in winter and summer was set as 12 L/(m^2^·h) and 24 L/(m^2^·h), respectively. According to the step-wise method, critical flux was defined as the flux above which the increase in TMP exceeded 0.4 kPa in one step duration (15 min) [23]. The mixed liquor suspended solids (MLSS) concentration in the A/O-MBR was about 6 g/L and the specific aeration demand (SAD_m_) was 1.0 m^3^/(m^2^·h).

#### 2.5.4. Fouling Rate

The fouling rates of the membranes were determined in the pilot-scale submerged A/O-MBR under a constant ultrahigh operation flux of 60 L/(m^2^·h) over 30 min. TMP was recorded every 5 min. Membrane fouling potentials were reflected by the changes in the membrane resistance within 30 min [24]. The fouling rate was also measured both in winter and summer at the same time of the critical flux measurement process.

#### 2.5.5. Filtration Process in A/O-MBR

In order to comprehensively evaluate the antifouling performance of PVDF/TiO_2_ composite membranes E1–E5, the long-term filtration process was actually carried out in the A/O-MBR targeting the municipal wastewater treatment in Shanghai. The filtration process was operated under the constant permeate flux of 20 L/(m^2^·h) and the intermittent suction ratio of 10 min: 2 min. The aqueous temperature of the process was 10 °C and the MLSS was about 6 g/L. TMP was recorded daily. The experiment was considered to have ended when the TMP of any membrane reached around 30 kPa.

## 3. Results and Discussion

### 3.1. TiO_2_ Nanoparticles Dispersion

Figure 1 shows the dispersion of TiO_2_ nanoparticles in solutions with DMSO and various contents of PEG. The TiO_2_ nanoparticles in the suspension for preparing PVDF/TiO_2_ composite membrane E1 exhibited the most serious agglomeration, which was followed by the suspensions corresponding to PVDF/TiO_2_ composite membranes E2, E3, E5 and E4. The most serious agglomeration shown in the suspension for membrane E1 might be attributed to the severe attraction interaction of the hydrophilic TiO_2_ nanoparticles with the high surface energy. Moreover, as supposed in our previous study, with the existence of the non-ionic surfactant PEG for membranes E2–E5, flower-like micelles might be formed owing to the hydrophilic interaction between the hydroxyl group on PEG and the TiO_2_ nanoparticles, which might alleviate the agglomeration of TiO_2_ nanoparticles as a result [11]. The even dispersion of TiO_2_ nanoparticles in the suspension for membrane E4 might be attributed to the proper steric hindrance of 6 wt.% PEG. In the suspension for membrane E5, the obvious agglomeration of TiO_2_ nanoparticles might have reduced due to the adhesion of excessive PEG chains. TiO_2_ nanoparticles were distributed evenly on the surface of copper wire mesh for the TEM determination of E4, while severely agglomerated TiO_2_ nanoparticles were only found in some specific areas of the wire mesh for E3 and E5, which resulted in fewer TiO_2_ nanoparticles in the suspension for membrane E4 being found per unit of image area than that for E3 and E5, as shown in Figure 1.

Figure 2 shows the values of *TSI*, which imply the dynamic dispersion status of the TiO_2_ nanoparticles in the composite casting solutions for membranes E1–E5. The highest *TSI* of the composite casting solution for membrane E1 suggested the lowest stability without the presence of the surfactant PEG. The lowest *TSI* value (<1.0) throughout the measurement period revealed the most stable dispersion of TiO_2_ nanoparticles in the composite casting solution for membrane E4, which might be ascribed to the preferable steric hindrance interaction under the harmonious proportion between the TiO_2_ nanoparticles and PEG. *TSI* for membranes E2 and E3 indicated that the stability of the composite casting solution was obviously improved with the increase in PEG content. However, the improvement diminished when the PEG content was over 6 wt.%. Therefore, 6 wt.% might be the desirable dosage of PEG for preparing the PVDF/TiO_2_ composite membrane in this study.

### 3.2. Membrane Characterizations

Figure 3 shows the surface and cross-sectional morphologies of PVDF/TiO_2_ composite membranes E1–E5. It can be observed in Figure 3a that all the membranes had similar surface morphologies, and the pore sizes of PVDF/TiO_2_ composite membrane E4 were obviously larger than those of the others. Figure 3b shows the cross-sectional morphologies of homogeneous composite membranes E1–E5. As shown in Figure 3b, the finger-like macrovoids were most fully developed in PVDF/TiO_2_ composite membrane E4 but were confined in membrane E5. According to the kinetics of film formation, the development of macrovoids can be significantly affected by the exchange velocity between the solvent and non-solvent phases during the immersing phase inversion process [25]. PEG was concurrently involved as a pore-forming additive in the phase inversion process. The fully developed finger-like structure in membrane E4 might be attributed to the high precipitation rate, which might have resulted from the rational matching of PEG and the TiO_2_ nanoparticles [26]. PEG might be rapidly exchanged to the coagulation bath after immobilizing TiO_2_ nanoparticles in the three-dimensional network of the membrane under the influence of the steric hindrance effect. However, for membrane E5, the entanglement between TiO_2_ nanoparticles and excessive PEG chains might slow down the precipitation rate during the phase inversion process in the coagulation bath, resulting in the dense spongy-like structure in the cross-section. Moreover, the considerable number of PEG chains adhered to TiO_2_ nanoparticles might also lead to the increase in the viscosity of the composite casting solution, which might also hinder the development of macrovoids [27].

The thicknesses of asymmetric PVDF/TiO_2_ composite membrane E1–E5 with fabric supporting layers are shown in Table 1. The thickness of membrane E5 as recorded in Table 1 was slightly increased owing to the slow exchange process between the solvent and non-solvent phases in the coagulation bath. Compared to the other membranes, the average pore size on the surface of PVDF/TiO_2_ composite membrane E4 was significantly increased to 0.129 μm. The porosity of PVDF/TiO_2_ composite membrane E5 decreased, which agrees with the suppressed development of macrovoids shown in Figure 3b. The water permeability of PVDF/TiO_2_ composite membrane E4 was evidently improved, which might be ascribed to the enhanced surface pore size and fully developed finger-like structure in the sub-layer. Conversely, the spongy-like structure might critically impede the filtration of water, resulting to the undesirable water permeability of membrane E5.

The mechanical properties of PVDF/TiO_2_ composite membranes E1–E5 estimated in terms of tensile strength and elongation at break are indicated in Figure 4. It can be observed that the tensile strength of PVDF/TiO_2_ composite membranes E2–E5 were reinforced by the involvement of PEG. Compared to membrane E1 without PEG, the agglomeration of the TiO_2_ nanoparticles in membranes E2–E5 might have diminished, and the TiO_2_ nanoparticles were more evenly dispersed in the polymeric network throughout the membrane bulk under the steric hindrance interaction of PEG. The TiO_2_ nanoparticles acting as cross-linking points might have intensified the interaction of the polymeric chains in membranes E2–E5, suggesting that more energy was needed to conquer the interaction or break down the bond between them, thus increasing the tensile strength [28]. Even the dispersion and wild agglomeration of TiO_2_ nanoparticles might have provided abundant cross-linking points and consequently resulted in strong interactions throughout the membrane bulk. The reason for the highest tensile strength for PVDF/TiO_2_ composite membrane E4 might be attributed to the even distribution of TiO_2_ nanoparticles in the membrane under the steric interaction of the optimal proportion of PEG. The springless spongy-like structure shown in Figure 3b might have been the reason for the low elongation at break for membrane E5.

The ATR–FTIR spectra for PVDF/TiO_2_ composite membranes E1–E5 are shown in Appendix A and Figure 5. The peak at 1400 cm^−1^ was associated with the deformation vibration of -CH_2_ [29]. The peaks at 1275 cm^−1^ and 1178 cm^−1^ were associated with the symmetrical and asymmetrical stretching of -CF_2_, respectively. The peak at 1065 cm^−1^ was assigned as the stretching vibration of -OH. The peaks at 875 cm^−1^ and 840 cm^−1^ were attributed to one of the characteristic peaks of PVDF and the stretching vibration of -CH, respectively. Hydrophilic flower-like micelles formed by the hydrophilic interaction between the hydroxyl group on PEG and the TiO_2_ nanoparticles might have benefited the stable dispersion of the TiO_2_ nanoparticles in the casting solutions and their immobilization on the membrane surfaces. This might have resulted from the fact that hydrophilic flow-like micelles tend to make contact with water (non-solvent phase in this study) during the phase exchange process in the coagulation bath. No obvious peak around 3400 cm^−1^ associated with the stretching vibration of Ti-OH was found in Appendix A, implying that PEG had been exchanged to the coagulation bath and was absent in the prepared composite membranes. The weakest signal of -OH observed in membrane E1 probably demonstrated the lowest distribution of TiO_2_ nanoparticles on the surface. The strong signal of -OH observed in membrane E4 might have been due to the uniform distribution of TiO_2_ nanoparticles on the surface. The signal of -OH was not positively strengthened with the increase in PEG content for membrane E5. This might be owing to the conjecture that fewer TiO_2_ nanoparticles were distributed on the membrane surface under the entanglement of excessive PEG chains and TiO_2_ nanoparticles in the sub-layer, as shown in Figure 3b.

The contact angle and Zeta potential of PVDF/TiO_2_ composite membranes E1–E5 are noted in Table 2. Compared to membrane E1, the negative Zeta potential and hydrophilicities characterized by the contact angle of membranes E2–E5 were evidently enhanced owing to the distribution of TiO_2_ nanoparticles under the steric hindrance effect of PEG during the film formation process. The relatively high hydrophilicity of membrane E4 contributed to the improvement of water permeability (as noted in Table 1). The 3D images and roughness of PVDF/TiO_2_ composite membranes E1–E5 determined by AFM are displayed in Figure 6. PVDF/TiO_2_ composite membranes E2–E5 exhibited lower roughness than that of membrane E1 without PEG, which might have been due to the amendatory agglomeration under the effect of the steric hindrance of PEG. The lowest roughness of membrane E4 illustrated by the lowest *R*a and *R*q indicated the minimum contact sites for foulants, hinting at the superior antifouling performance.

### 3.3. Assessment of Membrane Antifouling Performance

As shown in Table 2, the TOC concentration and Zeta potential of SMP were about 11.4 mg/L and −10.2 mV, respectively. The size of the SMP determined by using the Zetasizer analyzer was 443.3 ± 13.9 nm in this study. The contact angle of the SMP and PVDF/TiO_2_ composite membranes E1–E5 determined by utilizing three probe liquids are also displayed in Table 2. The surface tension parameters for each membrane are displayed in Table 3. The average surface tension parameters (γ^LW^, γ^+^, γ−) of the SMP used in the adhesion calculation were 44.03 mJ/m^2^, 0.01 mJ/m^2^ and 10.97 mJ/m^2^, respectively. The free energy of cohesion of each membrane calculated by adapting surface tension parameters in Equations (7)–(9) implied the attraction tendency of the membrane itself. The free energy of adhesion between each membrane and the SMP indicated the interaction tendency between the membrane surface and SMP. Δ*G*^EL^ was generally negligible and the negative value of Δ*G*^TOT^ could represent the attraction force between the membrane surface and SMP in the bulk [21]. A higher negative value suggests a stronger attraction tendency of foulants to the membrane surface, which means a severe membrane fouling property. Conversely, a lower negative value of free energy of cohesion or adhesion indicates a lower membrane fouling tendency. As shown in Table 3, the lowest negative value of cohesion and adhesion of membrane E4 signified the lowest attraction tendency for foulants to membrane surface and the best antifouling potential. This result might be attributed to the abundant hydroxyl groups of uniformly distributed TiO_2_ nanoparticles on the surface of membrane E4.

Figure 7a shows the interaction energy between membrane surfaces and SMP when foulants approach. The positive value of the interaction energy denoted the repulsive interaction between membrane surfaces and the approaching foulants. The interaction energy of membranes E1–E5 in ascending order was 0.36 KT (membrane E1), 0.41 KT (membrane E2), 0.44 KT (membrane E3), 0.48 KT (membrane E5) and 0.65 KT (membrane E4). The interaction energy peak for membrane E4 was obviously higher than those of membranes E1–E3 and E5, indicating that higher energy was needed for the foulants to be attached to the membrane surface. The progressively increased interaction energy of the PVDF/TiO_2_ composite membranes revealed the gradually enhanced antifouling performance. The relative flux shown in Figure 7b verified the theoretical prediction of antifouling performance for membranes E1–E5. In the SMP filtration experiment, membrane E4 exhibited the lowest flux declination, which might have been due to the improved hydrophilicity and permeability. The decreased roughness might also have benefited the enhanced antifouling performance of membrane E4 by reducing the contact site and the deposition of foulants on the membrane surface [30].

The critical flux defined a condition that the hydrodynamic drag force transporting colloids from the bulk to the membrane surface was roughly balanced with repulsive interaction forces [31]. According to the critical flux hypothesis, severe fouling might not be observed during the initial stage of operation under the critical flux [2]. Once the operating flux exceeds the critical flux, membrane fouling and the rate of filtration resistance accelerates [32]. Therefore, critical flux was generally used to guide the determination of operation flux in the application of membranes in MBR. In order to precisely investigate the antifouling performance of PVDF/TiO_2_ composite membranes E1–E5 in MBR, critical fluxes were determined both in summer and winter with the aqueous temperature of 14 °C and 24 °C, respectively. As shown in Figure 8a, the critical flux of membranes E2–E5 were variously increased by the effect of additive PEG compared to membrane E1. The critical flux of membrane E4 was enhanced the furthest to 36.8 L/(m^2^·h) in summer and 22.8 L/(m^2^·h) in winter, which indicated an excellent antifouling performance in MBR. In the same operation condition of operation flux, the fouling of membrane E4 might be the weakest. As observed in Figure 8b, the fouling rates conducted simultaneously in the same A/O-MBR device exhibited a similar fouling tendency, which was in accordance with the critical flux results. PVDF/TiO_2_ composite membranes E2–E5 indicated better antifouling performance than membrane E1, and membrane E4 exhibited the optimal antifouling performance with the lowest fouling rate both in summer and winter. Targeting the application in municipal wastewater treatment, an actual running operation in A/O-MBR was especially carried out in winter with the aqueous temperature of 10 °C. As indicated in Figure 9, the TMP of membrane E1 first reached the running end of 30 kPa, indicating the fastest fouling behavior. The TMP of membrane E4 was only 19.3 kPa followed by membranes E3, E5 and E2, suggesting the best antifouling performance. As a consequence, membrane E4 was recognized as the optimal and suitable membrane material in the municipal wastewater treatment application. The reason for the excellent antifouling performance of membrane E4 might be attributed to the fact that under the steric hindrance effect of PEG with the content of 6 wt.%, TiO_2_ nanoparticles were stably dispersed in the composite casting solution and uniformly immobilized on the membrane surface during the film formation process, thereby improving hydrophilicity, Zeta potential and roughness, etc. The fully developed finger-like structure also contributed to the improvement of the antifouling performance.

## 4. Conclusions

In this study, the influence of the content of PEG, simultaneously serving as the dispersing surfactant of TiO_2_ nanoparticles in suspensions and the pore-forming additive during the subsequent membrane fabrication process, was systematically investigated. The slight overlapping morphology of the TiO_2_ nanoparticles in suspension and the low *TSI* value (<1) manifested excellent dispersion with 6 wt.% PEG via steric interactions. The optimal content of PEG, a large, uniform pore size and the fully-developed finger-like structure on the sub-layer of the PVDF/TiO_2_ composite membrane were accomplished, which in turn improved porosity and water permeability. The mechanical properties, Zeta potential and hydrophilicity were also enhanced. The pre-dispersion of the TiO_2_ nanoparticles under the steric hindrance interaction of PEG also contributed to the decrease in roughness on the membrane surfaces demonstrated by 3D morphology and the low values of Ra and Rq. The decreasing negative free energy of adhesion (about −42.87 mJ/m^2^)) and increasing interaction energy (up to 0.65 KT) between the membrane surfaces and SMP, along with the less declined flux, confirmed the excellent modification effect on antifouling performance. The high critical flux and low fouling rate both in winter and summer verified the preferable antifouling performance in A/O-MBR. The lowest TMP (19.3 kPa) in the long-term running operation further verified the acceptable antifouling property in the municipal wastewater treatment application.

## Figures and Tables

**Figure 1 membranes-12-01118-f001:**
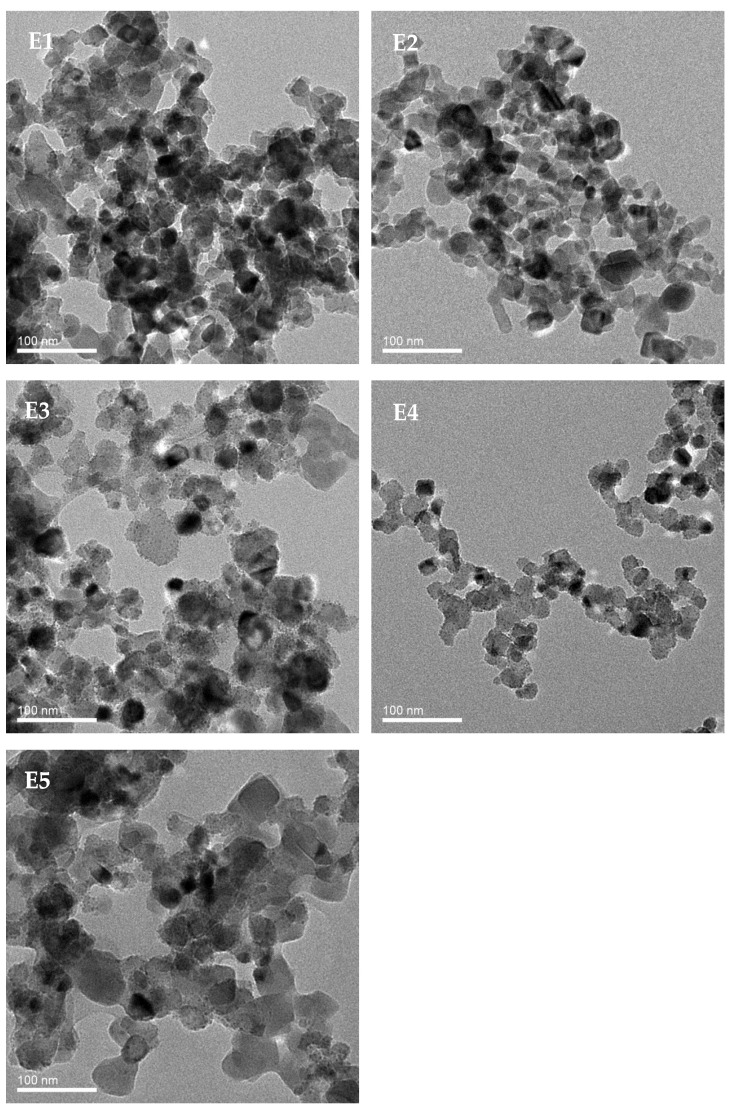
TEM images for TiO_2_ nanoparticles dispersed by DMSO and various contents of PEG.

**Figure 2 membranes-12-01118-f002:**
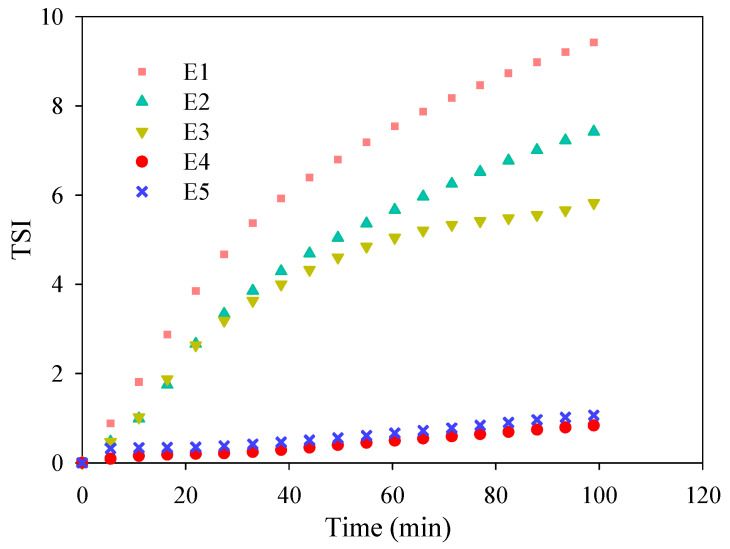
*TSI* of casting solutions of PVDF/TiO_2_ composite membranes E1–E5 throughout the measurement period of 100 min.

**Figure 3 membranes-12-01118-f003:**
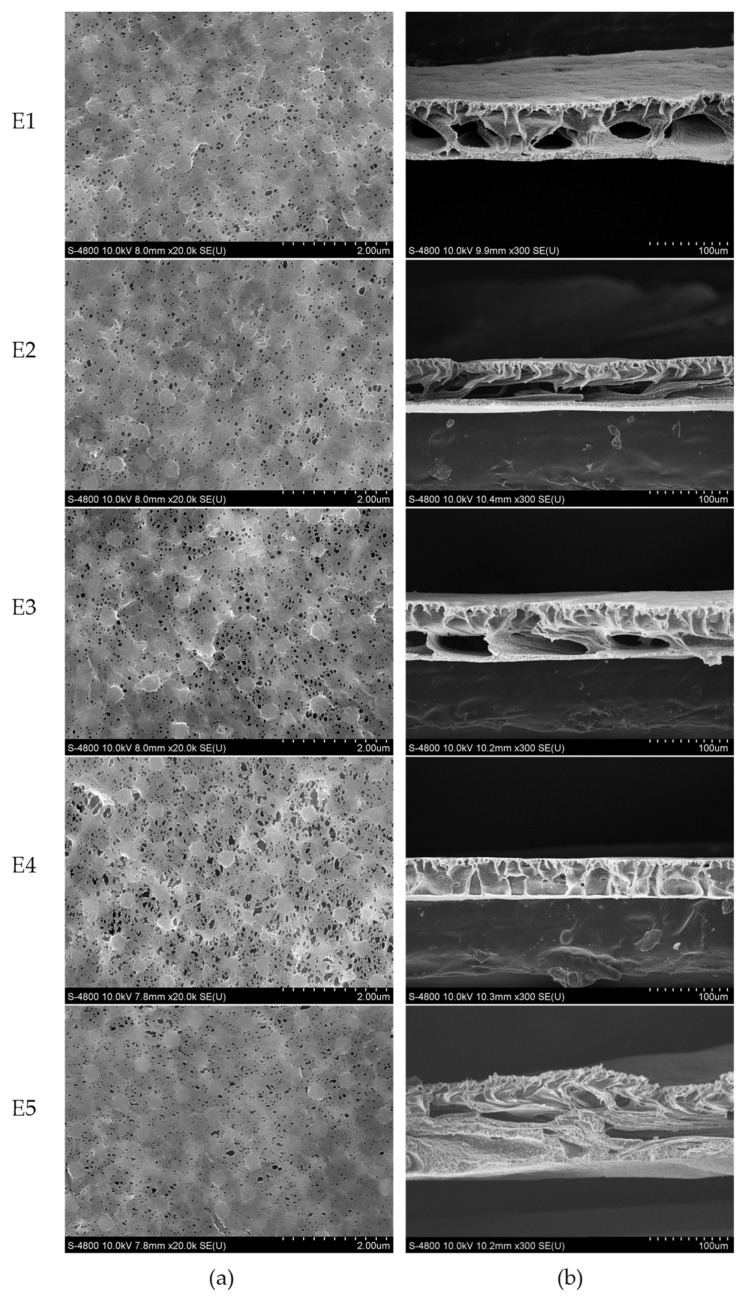
Morphologies of PVDF/TiO_2_ composite membranes E1–E5: (**a**) surface morphologies and (**b**) cross-sectional morphologies.

**Figure 4 membranes-12-01118-f004:**
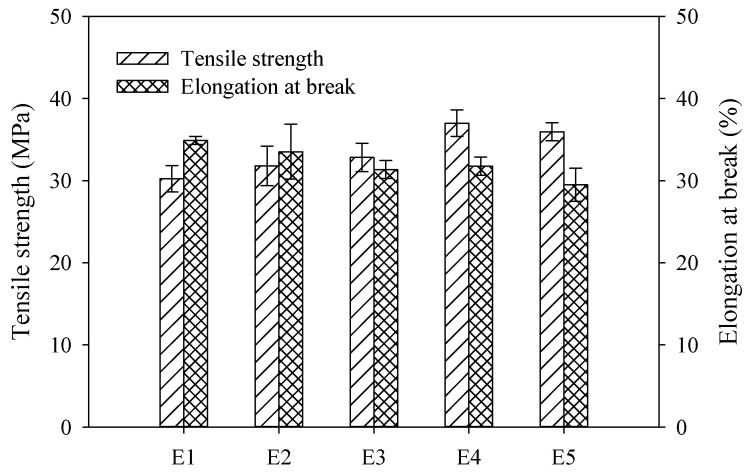
Mechanical strength of PVDF/TiO_2_ composite membranes E1–E5.

**Figure 5 membranes-12-01118-f005:**
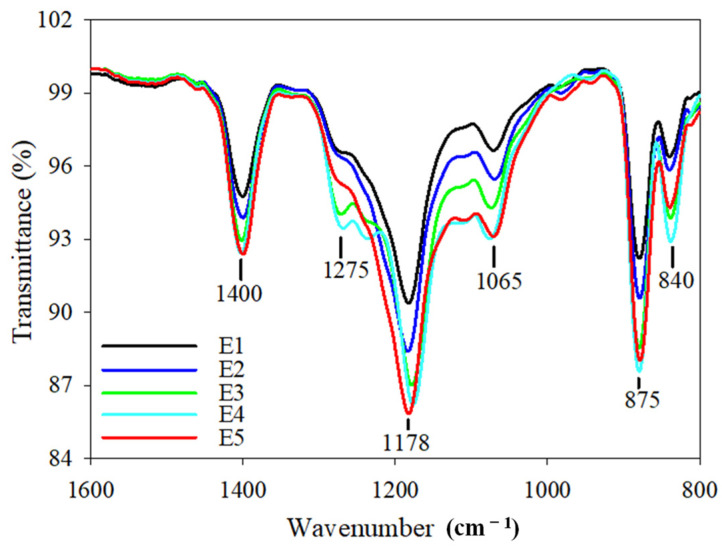
ATR–FTIR spectra of PVDF/TiO_2_ composite membranes E1–E5.

**Figure 6 membranes-12-01118-f006:**
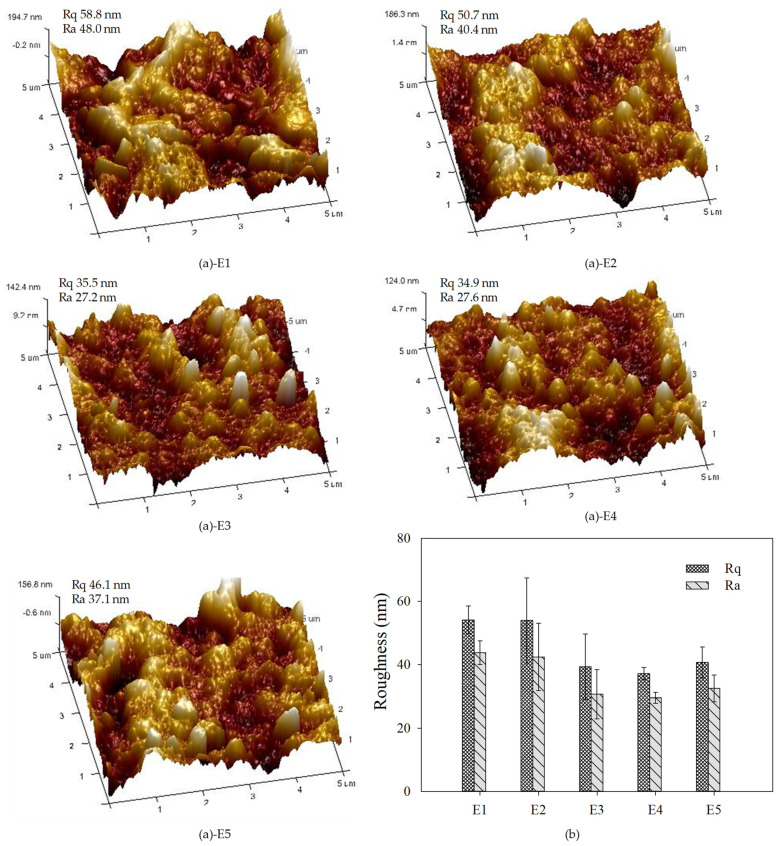
(**a**) Three-dimensional morphology and (**b**) roughness of PVDF/TiO_2_ composite membranes E1–E5 determined by AFM.

**Figure 7 membranes-12-01118-f007:**
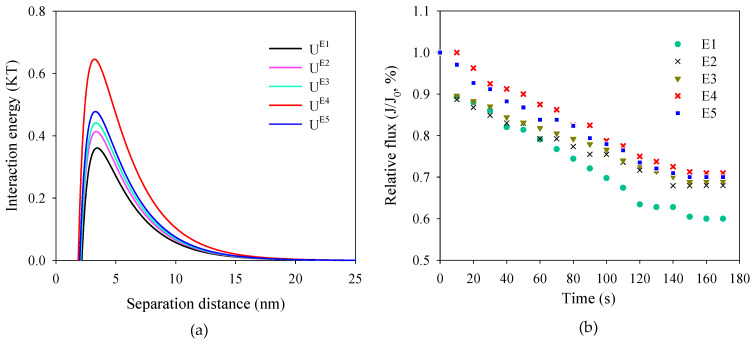
(**a**) Interaction energy between membrane surfaces and approaching foulants and (**b**) normalized flux of membranes when filtrating SMP solution.

**Figure 8 membranes-12-01118-f008:**
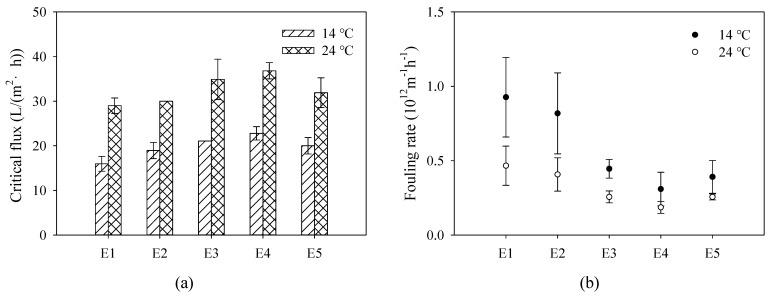
(**a**) Critical flux and (**b**) fouling rate of PVDF/TiO_2_ composite membranes E1–E5 determined both in winter (14 °C) and summer (24 °C) in A/O-MBR.

**Figure 9 membranes-12-01118-f009:**
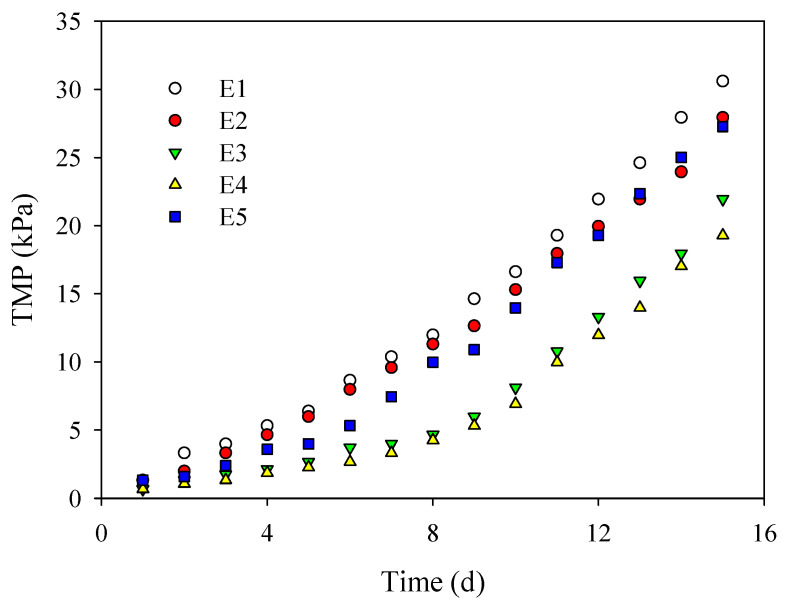
TMP of PVDF/TiO_2_ composite membranes E1–E5 running in A/O-MBR in winter (10 °C).

**Table 1 membranes-12-01118-t001:** Properties of the PVDF/TiO_2_ composite membranes dispersed by various contents of PEG (*n* = 3).

Membrane No.	Thickness (mm)	Average Pore Size (μm)	Porosity (%)	Water Permeability (L/(m^2^·h·kPa)
E1	0.24 ± 0.00	0.067 ± 0.012	43.4 ± 4.1	58.5 ± 1.3
E2	0.25 ± 0.00	0.072 ± 0.009	46.6 ± 3.5	58.1 ± 2.3
E3	0.24 ± 0.00	0.079 ± 0.009	45.2 ± 0.6	48.1 ± 1.8
E4	0.23 ± 0.00	0.129 ± 0.011	45.4 ± 2.2	71.5 ± 1.6
E5	0.26 ± 0.00	0.066 ± 0.011	40.1 ± 1.4	21.2 ± 2.4

**Table 2 membranes-12-01118-t002:** Properties of PVDF/TiO_2_ composite membranes E1–E5 and SMP (*n* = 3).

	**TOC Concentration (mg/L)**	**Zeta Potential (mV)**	**Contact Angle (°)**		
			**Water**	**Formamide**	**Diiodomethane**
SMP	11.4 ± 0.2	−10.2 ± 0.3	71.3 ± 4.1	51.7 ± 2.4	30.5 ± 0.9
**Membrane No.**	**PEG Content (%)**	**Zeta Potential (mV)**	**Contact Angle (°)**		
			**Water**	**Formamide**	**Diiodomethane**
E1	0%	−16.4 ± 0.7	87.2 ± 0.7	72.2 ± 2.6	62.1 ± 1.1
E2	2%	−18.6 ± 1.6	84.9 ± 0.3	69.5 ± 0.2	59.9 ± 0.4
E3	4%	−20.0 ± 1.9	81.5 ± 0.5	60.0 ± 0.4	55.5 ± 0.2
E4	6%	−30.0 ± 1.6	78.5 ± 0.3	57.4 ± 0.1	55.5 ± 0.5
E5	8%	−21.4 ± 0.8	80.7 ± 0.5	61.2 ± 1.0	56.5 ± 0.6

**Table 3 membranes-12-01118-t003:** Surface tension parameters and surface free energy of PVDF/TiO_2_ composite membranes at the separation distance of *h*_0_ (*n* = 3).

**Surface Tension Parameters for Each Membrane (mJ/m^2^)**						
Membrane No.	*γ* ^LW^	*γ* ^+^	*γ* ^−^		*γ* ^AB^	*γ* ^TOT^
E1	27.39 ± 0.61	0.07 ± 0.09	6.28 ± 0.64		1.05 ± 0.82	28.45 ± 1.37
E2	28.64 ± 0.26	0.08 ± 0.01	6.91 ± 0.20		1.45 ± 0.07	30.10 ± 0.19
E3	31.19 ± 0.14	0.67 ± 0.04	5.28 ± 0.22		3.76 ± 0.12	34.94 ± 0.24
E4	31.18 ± 0.27	0.88 ± 0.06	6.45 ± 0.33		4.77 ± 0.07	35.95 ± 0.19
E5	30.58 ± 0.32	0.55 ± 0.08	6.33 ± 0.21		3.71 ± 0.24	34.30 ± 0.49
	**The Free Energy of Cohesion of** **Membranes (mJ/m^2^)**			**The Free Energy of Adhesion of Membranes (mJ/m^2^)**		
Membrane No.	Δ*G*_121_^LW^	Δ_121_*G*^AB^	Δ*G*_121_^SWS^	Δ*G*_123_^LW^	Δ*G*_123_^AB^	Δ*G*_123_^SWS^
E1	−0.64 ± 0.13	−49.18 ± 0.65	−49.82 ± 0.75	−2.22 ± 0.23	−42.14 ± 0.67	−44.36 ± 0.86
E2	−0.93 ± 0.07	−46.24 ± 0.57	−47.17 ± 0.52	−2.69 ± 0.09	−40.69 ± 0.32	−43.38 ± 0.25
E3	−1.68 ± 0.05	−46.58 ± 0.81	−48.26 ± 0.76	−3.60 ± 0.05	−42.09 ± 0.47	−45.69 ± 0.42
E4	−1.68 ± 0.09	−41.27 ± 0.82	−42.95 ± 0.82	−3.60 ± 0.09	−39.27 ± 0.56	−42.87 ± 0.55
E5	−1.48 ± 0.10	−43.69 ± 0.37	−45.18 ± 0.29	−3.39 ± 0.11	−40.20 ± 0.26	−43.59 ± 0.23

## Data Availability

The data presented in this study are available on request from the corresponding author.

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
