# Peer review of "Influence of Dispersed TiO2 Nanoparticles via Steric Interaction on the Antifouling Performance of PVDF/TiO2 Composite Membranes"

_membranes, 2022, doi:10.3390/membranes12111118_

Round 1

Reviewer 1 Report

Decision:

Minor Revision

Comments

         The authors reported “Influence of Dispersed TiO 2  Nanoparticles via Steric Interaction on the Antifouling Performance of PVDF/TiO2  Composite” Membranes, this research article can be useful for the scientific community after some revision. Overall, the work is good and well-presented.  However, the authors should address the following points outlined below to improve scientific quality. After the suggested revisions are carefully addressed, this work may be considered for publication.

1.     The abstract is too long. Make it short with the important highlights of this work. It should be clear and informative with important highlights. The first few lines should be moved to the introduction section.

2.     Line 243 PVDF/TiO 2  composite membranes E1 exhibits the most serious agglomeration, Author should explain the reason for this agglomeration in detail.

3.     In AFM images write the values for the roughness for each sample.

4.     Did you check the XRD analysis

5.     Are the results reproducible? Kindly show data with % of reproducibility.

6.     There are many grammatical errors, recheck the manuscript once again for all typo errors.

Author Response

Response to Reviewer 1 Comments

The authors reported “Influence of Dispersed TiO2 Nanoparticles via Steric Interaction on the Antifouling Performance of PVDF/TiO2 Composite” Membranes, this research article can be useful for the scientific community after some revision. Overall, the work is good and well-presented.  However, the authors should address the following points outlined below to improve scientific quality. After the suggested revisions are carefully addressed, this work may be considered for publication.

Response: Thank you for the positive comment. According to your comments, we have made some modifications to strengthen the substance of the manuscript. We hope these revisions will meet with your approval.

Point 1:The abstract is too long. Make it short with the important highlights of this work. It should be clear and informative with important highlights. The first few lines should be moved to the introduction section.

Response 1: Thanks a lot for the nice advice.

The first sentence “Incorporation of polymeric membrane and functional hydrophilic TiO2 nanoparticles has attracted much attention to deal with the thorny fouling problem.” in the abstract has been removed and the corresponding description in the introduction section has been revised as Line 44-46 on Page 1.

    Line 44-46 on Page 1:

membrane modification by incorporating original membrane material with nanoparticles has been identified as the straightforward way to deal with the thorny membrane fouling issue”.

The second sentence “The dispersion mechanism of a small amount of TiO2 nanoparticles by varying content of organic surfactant along with the influence on the antifouling modification of polymer/TiO2 composite membrane has rarely been highlighted.” In the abstract has been removed and the corresponding information has already been described in Line 73-76 in the introduction section.

Line 73-76 on Page 2:

In view of the detrimental influence of superfluous nanoparticles in sewage treatment system, the dispersion mechanism of a small amount of TiO2 nanoparticles by varying content of organic surfactant deserves a clearly consideration in the antifouling performance modification of polymer/TiO2 composite membrane.

Point 2:Line 243 PVDF/TiO2 composite membranes E1 exhibits the most serious agglomeration, Author should explain the reason for this agglomeration in detail.

Response 2: Thank you very much for the valuable advice. The explanation for Line 239 in the revised version has been added in Line 244-247, and the following explanation for membranes E2-E5 has been revised as Line 247-250.

Line241-247 on Page 6:

“The most serious agglomeration shown in the suspension for membrane E1 might be attributed to the severe attraction interaction of hydrophilic TiO2 nanoparticles with high surface energy. Moreover, as supposed in our previous study, with the existence of the non-ionic surfactant PEG for membranes E2-E5, flower-like micelles might be formed owing to the hydrophilic interaction between the hydroxyl on the PEG and the TiO2 nanoparticles, which might alleviate the agglomeration of TiO2 nanoparticles as a result [11].

Point 3:In AFM images write the values for the roughness for each sample.

Response 3: Thanks a lot for the advice. Information of roughness (Rq and Ra) for each membrane has been added in Figure 6 on Page 12.

Figure 6 on Page 12:

“Figure 6. (a)Three-dimensional morphology and (b) roughness of PVDF/TiO2 composite membranes E1-E5 determined by AFM.”

Point 4:Did you check the XRD analysis?

Response 4: We appreciate your valuable advice very much. We have not paid much attention on the XRD analysis in our research and we will take the characterization into consideration to insight the influence of TiO2 nanoparticles thoroughly in our future study. Thanks a lot for the helpful information again.

Point 5:Are the results reproducible? Kindly show data with % of reproducibility.

Response 5: Thank you very much for the comment. Results in our study are reproducible and have been repeatedly measured for three times, shown as “n=3” in Table 1 on Page 6. Results are recorded as “average value ± standard deviation”. Results shown in figures are also plotted based on the way of expression.

Point 6:There are many grammatical errors, recheck the manuscript once again for all typo errors.

Response 6: Thanks a lot for the comment. We feel sorry for our weak writings in the manuscript and the inconvenience they caused in your reading. We have checked and revised the manuscript to increase the readability, and these revisions did not list here but marked in red in the revised manuscript. We appreciate your careful reading very much.

Reviewer 2 Report

This manuscript is interesting but it should be revised to publish.

·         The work deals with soluble microbial products (SMP) as foulants. However, any information on the SMP nature is absent in the Introduction. It is suggested to be added information about the SMP composition, size, and chemical or other activity, which can lead to membrane fouling.

·         According to Fig. 1, suspension for membrane E4 contains considerably less material per unit of image area, then that for Е3 and Е5. Please add the explanation for it.

·         Page 9. The data on the thickness of the membranes in Table 1 do not correspond to the results on the thickness which can be seen in Fig. 3. Please check it and use adequate description.

·         Page 11. To prove “interaction between the hydroxyl on the PEG and the TiO2 nanoparticles”, a spectrum for a pair of PEG and TiO2 is suggested to be added in Fig. 5.

Author Response

Response to Reviewer 2 Comments

This manuscript is interesting but it should be revised to publish.

Response: Thanks a lot for the positive comment. According to your comments, we have made some modifications to strengthen the substance of the manuscript. Also, corresponding images with data of poor quality have been substituted in the revised version. We hope these revisions will meet with your approval.

Point1:The work deals with soluble microbial products (SMP) as foulants. However, any information on the SMP nature is absent in the Introduction. It is suggested to be added information about the SMP composition, size, and chemical or other activity, which can lead to membrane fouling.

Response 1: Thanks a lot for the helpful comment. The information has been added in the introduction section in Line 90-93 on Page 2. A new reference [13] has been added and the number of the following references have also been revised both in the main body and the reference section.

Line 90-93 on Page 2:

“In addition, SMP containing polysaccharides, protein, humic acid and other polymeric compounds, are excreted into solution during substrate metabolism, biomass growth, biomass decay, etc. and have been identified as the significant biological foulants [13].”

Reference section:

13.  Wang, Q.; Wang, Z.; Zhu, C.; Mei, X.; Wu, Z., Assessment of SMP fouling by foulant–membrane interaction energy analysis. J. Membr. Sci. 2013, 446, 154-163.

Point 2: According to Fig. 1, suspension for membrane E4 contains considerably less material per unit of image area, then that for Ð•3 and Ð•5. Please add the explanation for it.

Response 2: Thank you for the comment. The explanation has been added in Line 254-259 on Page 7.

Line 254-259 on Page 7:

TiO2 nanoparticles were distributed evenly on the surface of copper wire mesh for TEM determination of E4, while severe agglomerated TiO2 nanoparticles were only found in some specific area of wire mesh for E3 and E5, which caused the result that less TiO2 nanoparticles in the suspension for membrane E4 were found in per unit of image area than that for E3 and E5 as shown in Figure 1.

Point 3:Page 9. The data on the thickness of the membranes in Table 1 do not correspond to the results on the thickness which can be seen in Fig. 3. Please check it and use adequate description.

Response 3: Thanks a lot for the comment. The description of thicknesses in Figure 3 and Table 1 has been added respectively in Line 283-284 on Page 8 and Line 307-308 on Page 10.

Line 283-284 on Page 8 for Figure 3:

“Figure 3b shows the cross-sectional morphologies of homogeneous composite mem-branes E1-E5.”

Line 307-308 on Page 10 for Table 1:

“Thicknesses of asymmetric PVDF/TiO2 composite membrane E1-E5 with fabric supporting layers are shown in Table 1.”

Point 4:Page 11. To prove “interaction between the hydroxyl on the PEG and the TiO2 nanoparticles”, a spectrum for a pair of PEG and TiO2 is suggested to be added in Fig. 5.

Response 4: Thank you very much for the valuable comment. The full spectrum has been added in Figure S1 in the supporting information. The interaction between PEG and TiO2 worked during the film fabrication process in the coagulation bath, but absented in the fabricated membranes. The explanation of the interaction has been added in Line 346-349.

Line 346-349 on Page 11:

“No obvious peak around 3400 cm-1 associated with the stretching vibration of Ti-OH was found in Figure S1, implying that PEGs have been exchanged to the coagulation bath and were absent in the prepared composite membranes.”

Figure S1 in the supporting information:

“Figure S1. Full spectra of PVDF/TiO2 composite membranes E1–E5 determined by ATR-FTIR.”
